



# Potential for the measurement of MLT wind, temperature, density and geomagnetic field with Superconducting Submillimeter-Wave Limb-Emission Sounder-2 (SMILES-2)

Philippe Baron[1], Satoshi Ochiai[1], Eric Dupuy[2], Richard Larsson[3], Huixin Liu[4], Naohiro Manago[5], Donal Murtagh[6], Shin-ichiro Oyama[7,8,9], Hideo Sagawa[10], Akinori Saito[11], Takatoshi Sakazaki[11], Masato Shiotani[11], and Makoto Suzuki[12]

[1]National Institute of Information and Communications Technology, Koganei, Japan
[2]National Institute for Environmental Studies, Tsukuba, Japan
[3]Max Planck Institute, Göttingen, Germany
[4]Kyushu University, Kyushu, Japan
[5]Chiba University, Chiba, Japan
[6]Chalmers University of Technology, Göteborg, Sweden
[7]Nagoya University, Nagoya, Japan
[8]National Institute of Polar Research, Tachikawa, Japan
[9]University of Oulu, Oulu, Finland
[10]Kyoto Sangyo University, Kyoto, Japan
[11]Kyoto University, Kyoto, Japan
[12]Japan Aerospace Exploration Agency, Sagamihara, Japan

*Correspondence to:* P. Baron (baron@nict.go.jp)

**Abstract.** Submillimeter-Wave Limb-Emission Sounder-2 (SMILES-2) is a satellite mission proposed in Japan to probe the middle and upper-atmosphere (20–160 km). The main instrument is composed of 4-K cooled radiometers operating near 0.7 and 2 THz. It could measure the diurnal changes of the horizontal wind above 30 km, temperature above 20 km, ground-state atomic oxygen above 90 km, atmospheric density near the mesopause, as well as abundance of about 15 chemical species. In

this study we have conducted simulations to assess the wind, temperature and density retrieval performance in the mesosphere and lower thermosphere (60–110 km) using the radiometer at 760 GHz. It contains lines of water vapor ($H_2O$), molecular oxygen ($O_2$) and nitric oxide (NO) that are the strongest signals measured with SMILES-2 at these altitudes. The Zeeman effect on the $O_2$ line due to the geomagnetic field ($\boldsymbol{B}$) is considered, otherwise, the retrieval errors would be underestimated by a factor of 2 above 90 km. The optimal configuration for the radiometer's polarization is found to be vertical linear. The

line-of-sight wind is retrieved with a precision of 2–5 $\mathrm{m\,s^{-1}}$ up to 90 km (30 $\mathrm{m\,s^{-1}}$ at 110 km) and a vertical resolution of 2.5 km. Temperature and atmospheric density are retrieved with a precision better than 5 K (30 K) and 7% (20%) up to 90 km (110 km), respectively. Errors induced by uncertainties on the vector $\boldsymbol{B}$ are mitigated by retrieving it. The retrieval of $\boldsymbol{B}$ is described as a side-product of the mission. At high-latitudes, precisions of 30–100 nT on the vertical component and 100–300 nT on the horizontal one could be obtained at 85 and 105 km (vertical resolution of 20 km). SMILES-2 could therefore provide

the first measurements of $\boldsymbol{B}$ close to the electrojets' altitude, and the precision is enough to measure variations induced by solar storms in the auroral regions.



# 1 Introduction

The mesosphere and lower thermosphere (MLT) is a transitional region (60–110 km) between atmospheric layers with very different characteristics, namely the stratosphere (15–60 km) and the thermosphere (90–400 km) (Smith, 2012; Shiotani et al., 2019). In the stratosphere, $O_3$ controls the chemical and radiative processes, hence it also regulates the temperature and the

dynamics. In the thermosphere, the chemistry and the radiative balance are mainly controlled by the oxygen atoms. In this region, wind and temperature exhibit large diurnal variations and are strongly influenced by tides generated in the lower atmosphere. The thermosphere is also the region of interactions between the ionized (plasma) and neutral atmosphere.

The mean physical characteristics of the MLT (wind, temperature and density) are primarily established by energy transfered from the troposphere via small-scale gravity waves (GWs) (Fritts and Alexander, 2003; Tsuda, 2014). Hence, the MLT state

deviates significantly from the radiative equilibrium as illustrated by the occurrence of the coldest point of the Earth system ($\approx 150$ K) in the summer polar mesopause. Waves with planetary scales also contribute to the upper atmosphere climate (general circulation) through their momentum and energy transport/deposition (Forbes et al., 2006; Pancheva and Mukhtarov, 2011). In particular, tides that are mainly driven by diurnally varying diabatic heating in the troposphere and the stratosphere, propagate upward, with their amplitude reaching a maximum in the MLT (Chapman and Lindzen, 1970; Sakazaki et al., 2015).

Hence, the MLT plays a key role in connecting the lower and upper atmosphere and also in linking both hemispheres (Xu et al., 2009; Karlsson and Becker, 2016). Furthermore, the increase of anthropogenic $CO_2$ is responsible for a cooling of 1–3 K/decade in the MLT that has been measured since the early 1990s (Beig, 2011).

The processes behind these phenomena are still not well quantified. The difficulty arises from the non-linear interactions between the GWs, tides, planetary waves, the background wind and the electromagnetic field (Sato et al., 2018; Immel et al.,

2006). The system is further complicated by the interconnections between the dynamics and highly variable chemical species, as well as the very different temporal and spatial scales of these processes. Observations of the MLT, in particular of wind, temperature and density, are therefore essential to further our understanding of this region (Smith, 2012).

Continuous measurements of temperature and wind are performed from ground-based stations using lidars (Steinbrecht et al., 2009; Baumgarten, 2010), radars (Jacobi et al., 2015; Tsutsumi et al., 2017) and, up to 70 km, with millimeter radiome-

ters (Rüfenacht et al., 2014). Density was recently monitored using meteor radars (Yi et al., 2018) but measurements remain scarce. Satellite observations of the MLT have also been performed for several decades. The missions currently in operation and capable of measuring at these altitudes are listed in Tab. 1. Temperature is measured with various techniques and spectral domains (Schwartz et al., 2006; Sica et al., 2008; Sheese et al., 2010; Christensen et al., 2015; Eastes et al., 2017; Englert et al., 2017), but discrepancies larger than 10 K can be found between these measurements above 80 km (García-Comas et al.,

2014). Baron et al. (2013) and Shepherd (2015) described the past and current horizontal wind measurements from space. Currently only TIDI and MLS (and soon MIGHTI) are capable of measuring MLT winds but with a poor sensitivity below 80 km (Niciejewski et al., 2006; Wu et al., 2008; Englert et al., 2017).

In the future, we clearly risk a lack of satellite observations since all the current missions (except ICON) have already exceeded their theoretical lifetime. Sweden is preparing two Innosat-based missions that are of interest for the study of the



**Table 1.** Current and future satellites and instruments capable of measuring the MLT (60–110 km).

| Satellites in operation | Launch year |
| --- | --- |
| Odin | 2001 |
| Thermosphere Ionosphere Mesosphere Energetics and Dynamics, TIMED | 2001 |
| Atmospheric Chemistry Experiment, ACE | 2004 |
| Aura | 2004 |
| Ionospheric Connection Explorer, ICON | Second quarter of 2019 |
| Instruments | Main products |
| Sub-Millimeter limb sounder, SMR (Odin) | T, $H_2O$,NO |
| Optical Spectrograph and InfraRed Imaging System, OSIRIS (Odin) | T, airglow |
| Sounding of the Atmosphere using Broadband Emission Radiometry, SABER (TIMED) | T, O |
| TIMED Doppler Interferometer, TIDI (TIMED) | W, T, airglow |
| Microwave Limb Sounder, MLS (Aura) | T, (W) ; $< 90$ km |
| Michelson Interferometer for Global High-resolution Thermospheric Imaging, MIGHTI (ICON) | W, T, airglow ; $> 90$ km |
| Planned missions | |
| Mesospheric Airglow/Aerosol Tomography Spectroscopy, MATS (Innosat), 2019 | GWs,T,NLC ; 90–100 km |
| Stratospheric Inferred Winds, SIW (Innosat), 2023 | W, T ; $< 80$ km |

W, T and NLC denote temperature, wind and noctilucent cloud.

MLT (Tab. 1). The MATS mission aims at characterizing the 3D structure of the GWs near 90–100 km using the oxygen A-band emission and the ultraviolet light scattered by noctilucent clouds (Gumbel et al., 2018). Information on temperature will also be retrieved. The other mission is SIW, a sub-millimeter limb sounder that will measure horizontal wind, temperature and trace gases up to about 80 km (Baron et al., 2018). The MATS and SIW missions will be operational for 2 years between
2019–2021 and 2023–2025, respectively. Other projects have not been selected yet and remain uncertain. For example, Wu et al. (2016) proposed a THz limb sounder (TLS) to measure the atomic oxygen line at 2 THz. Such an instrument could fly together with a new version of SABER (Mlynczak and Yee, 2017). The European Space Agency (ESA) is studying a limb sounder operating between 0.8–4 THz for the retrieval of the abundance of chemical species such as atomic oxygen (O) or the hydroxyl radical (OH) (Gerber et al., 2013). Kaufmann et al. (2018) described a concept for a limb sounder onboard a cubeSat
to measure temperature with high horizontal resolution using the molecular oxygen ($O_2$) A-band infra-red emission.

Superconducting Submillimeter-Wave Limb-Emission Sounder 2 (SMILES-2) is a middle and upper atmospheric satellite mission proposed to the Japan Aerospace Exploration Agency (JAXA) (Ochiai et al., 2017, 2018; Shiotani et al., 2019). If selected, it will be launched around 2026 on a JAXA M-class satellite. The objectives are to provide geophysical information with unprecedented precision and altitude coverage such as the temperature between 15–160 km, horizontal wind between
30–160 km, atmospheric density up to 110 km, ground state of atomic oxygen between 90–160 km and more than 15 trace gases' abundance (Baron et al., 2019; Suzuki et al., 2019). The proposed satellite will be equipped with two antennas for the



limb measurement of horizontal winds, and three radiometers near 0.7 and 2 THz cooled at 4 K, a technology successfully tested with JEM/SMILES (Kikuchi et al., 2010). With a precessing orbit and the high receiver precision, it will be possible to retrieve diurnal variations of very weak signals as demonstrated with JEM/SMILES (Sakazaki et al., 2013; Khosravi et al., 2013).

In this study we discuss the potential for SMILES-2 to measure the main characteristics of the neutral MLT, namely wind, temperature and atmospheric density. An essential source of information is the $O_2$ transition at 773.8 GHz. As a magnetic dipole, $O_2$ is subject to the Zeeman effect induced by the Earth's magnetic field ($B$). Special care is taken to properly include this effect in the simulations in order to correctly assess the measurement performance. Retrieval errors induced by uncertainties on $B$ are mitigated by retrieving its three components simultaneously with other atmospheric parameters. The scientific interest

of the retrieval of $B$ is also discussed. In Sect. 2, the characteristics and principle of the observations are presented in details. Sections 3 and 4 describe the Zeeman model and the retrieval setting, respectively. The retrieval errors are discussed in Sect. 5. Finally, we summarize the results and discuss future analysis for SMILES-2.

## 2  Measurement principle

### 2.1  Observation method

The observation characteristics are summarized in Tables 2. The atmospheric limb is scanned from about 20 to 180 km. Scans are performed alternatively with two antennas looking at perpendicular directions to each other. Both antennas can probe the same atmospheric column with 7 min delay (Fig. 1), allowing us to derive the 2D horizontal winds. The same method will be used for SIW and more information is given in Baron et al. (2018). The limb geometry provides a high vertical resolution of 2–3 km, and the zonal and meridional samplings at the equator are about 20° (2200 km) and 6° (650 km), respectively.

The orbit precesses with a period of about 3 months. The satellite orientation is reversed after every half precession cycle in order to keep the solar panels properly illuminated and the radiative-cooling panels in the shadow side. The latitude coverage is between 50°S–80°N or 80 °S–50 °N depending on the satellite orientation. At low and mid latitudes, the same latitude is observed twice per orbit, with LT differences close to 12 hours. Hence, gathering the observations between each maneuver allows us to piece together the complete diurnal cycle of the retrieved parameters.

### 2.2  Spectral bands

Three spectral bands near 638 GHz, 763 GHz and 2 THz are measured simultaneously (Ochiai et al., 2018). The band at 638 GHz contains a strong stratospheric and lower mesospheric signal from ozone ($O_3$). This band is the same as that selected for SIW and its main characteristics are described in Baron et al. (2018). Two THz bands are measured alternatively, one contains OH lines and the second one an O line (Ochiai et al., 2017; Suzuki et al., 2019). The O line is used to retrieve between

90–160 km, the abundance of O in its ground-state, wind and temperature (Baron et al., 2015, 2019).



**Table 2.** SMILES2 observation characteristics.

| | |
|---|---|
| Satellite altitude | 550 km |
| Latitude range | 50°S–80°N or |
| | 80 °S–50 °N |
| Scan altitude | 20–200 km (4.27 °) |
| Scan velocity | 0.1 ° s$^{-1}$ (43 s/scan$^*$) |
| Spectrum integration time | 0.25 s (1.1 km at 80 km$^{**}$) |
| Antenna diameter | 75 cm |
| Frequency ⇔ velocity at 760 GHz | 1 m s$^{-1}$ ⇔ 2.5 kHz |

$^*$Calibration measurements will be performed over the upper-range.

$^{**}$ Tangent point vertical displacement during the integration time.

**Table 3.** The 763-GHz spectral band.

| Local oscillator (GHz) | Lower sideband (GHz) | Upper sideband (GHz) | DSB Tsys Resolution | Vertical resolution |
|---|---|---|---|---|
| 763.5 | 750.0–756.0 | 771.0–777.0 | 180 K | 0.0366 ° |
| | H$_2$O at 752 GHz | O$_2$ at 773 GHz | 0.5 MHz | 1.9 km$^*$ |

$^*$ Estimated for a tangent height of 80 km including the antenna FOV and the scan velocity.

The 763-GHz band (Table 3) is the band considered in this study. It contains lines of water vapor (H$_2$O) at 752.03 GHz and O$_2$ at 773.84 GHz (Fig. 2) that provide a strong signal in the MLT. It also contains other molecular lines, weaker but still suitable for our study: nitric oxide (NO, 751.67–752.00 GHz and 773.02–773.05 GHz), O$_3$ (754.46 GHz and 776.66 GHz) and carbon monoxide isotopologue ($^{13}$CO) at 771.183 GHz. The bands have changed compared to those originally described by Ochiai et al. (2017), a change motivated to reduce the power consumption. In the new setting, the CO line is about 50 times weaker than that previously selected.

### 2.3 Qualitative description of the information content

Most of the lines in the spectral bands are emitted by chemical species in their ground state under the local thermodynamic equilibrium. The molecular abundance and the temperature are retrieved from the amplitude of the lines. Their Doppler shift (2.5 kHz for 1 m s$^{-1}$) is used to retrieve the line-of-sight (LOS) wind. The atmospheric density is derived from the O$_2$ abundance considering that the volume mixing ratio of O$_2$ is well known below 110 km (Schwartz et al., 2006).

Above about 70 km, the lines are broadened by the random moelcular motions, i.e., Doppler broadening, and they do not carry direct information on the pressure (Appendix A). Consequently, the density of the molecule can be retrieved and not the





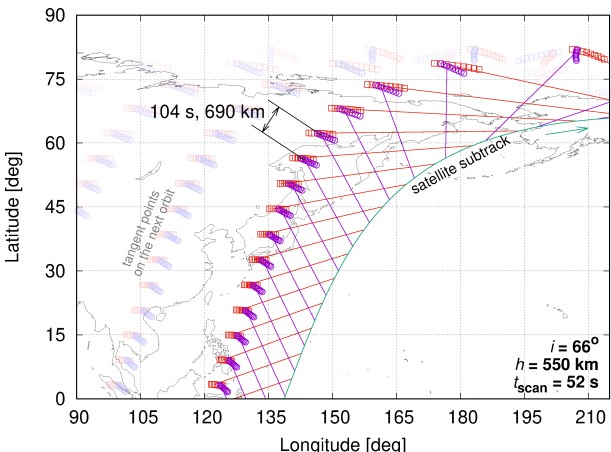

**Figure 1.** SMILES-2 orbit over the northern hemisphere. The red and purple lines show the forward and aftward line-of-sights (LOSs). The circles show the tangent-point footprints (Ochiai et al., 2017).

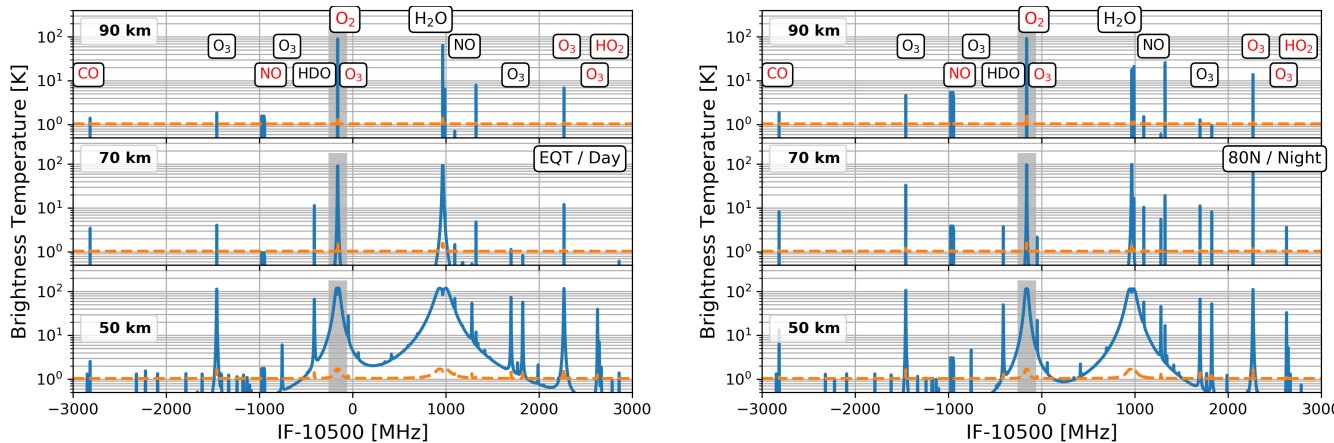

**Figure 2.** (a) Spectra at the Equator in daytime for tangent heights of 50, 70 and 90 km. The x-coordinates are the intermediate frequency (IF). The yellow dashed lines indicate the noise standard deviation (2-$\sigma$). The gray area shows the frequency range of 200 MHz in which the Zeeman radiative transfer model is used (only for the upper-side band). (b) Same as (a) but for 80°N and nighttime winter conditions. The red labels indicate molecular lines in the upper-side band. The LO frequency is 763.5 GHz.

volume-mixing ratio (VMR) as in the lower altitudes. The retrieval altitude is inferred using the star trackers and GPS aboard the satellite with a precision expected to be better than 100 m. A global altitude offset of the scan can be corrected using the pressure retrieved from the stratospheric lines.

The molecular oxygen is a magnetic dipole that interacts with $\boldsymbol{B}$. It is subject to the so-call Zeeman effect (Lenoir, 1968) and the selected spectroscopic transition is split into $\sigma^{\pm}$ and $\pi$ components with different polarization states depending on the LOS orientation (Fig. 3). The frequency separation of the spectral components is proportional to the amplitude of $\boldsymbol{B}$.





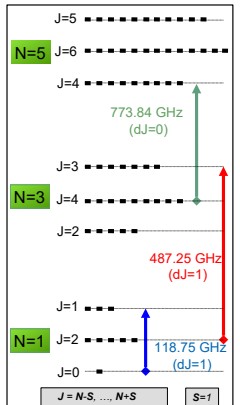 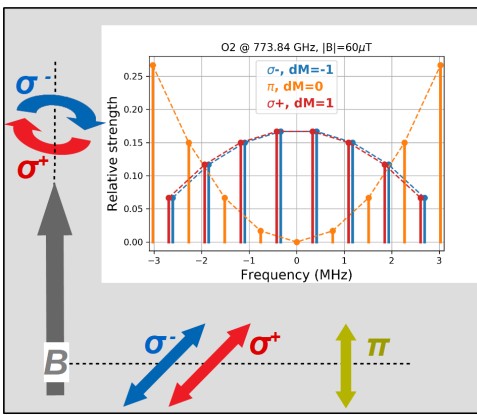

**Figure 3.** Left panel: Energy levels of the $O_2$ transitions measured with MLS at 119 GHz (Schwartz et al., 2006), SMR at 487 GHz (Larsson et al., 2014) and SMILES-2. The degenerated energy levels ($M=-J\ldots J$) are indicated with black horizontal strokes. Right panel: Strength and polarization of the Zeeman components of the line selected for SMILES-2. The components' frequency is computed for a magnetic field of 0.6 Gs. The representation of the polarization states is adapted from Fig. 3.1 in Landi Degl'Innocenti and Landolfi (2004). The dashed-lines represent perpendicular and parallel LOSs with respect to $B$ (gray thick arrow).

## 3 Zeeman effect modeling

The Zeeman effect on atmospheric molecular-oxygen lines has been extensively studied (Lenoir, 1968; Pardo et al., 1995; Schwartz et al., 2006; Larsson et al., 2014; Navas-Guzmán et al., 2015). In this study, we describe the polarized radiance with Stokes vectors as in Landi Degl'Innocenti and Landolfi (2004) (e.g., Eq. 1.32), Larsson et al. (2014) and Steiner et al. (2016).
5 The magnetic field characteristics (amplitude and orientation angles wrt the LOS) are defined at the LOS tangent height (Fig. 4) and are assumed constant over the LOS. This approximation is the same as that used by Yee et al. (2017) and it is justified since most of the retrieved information comes from a thin altitude range around the tangent point.

### 3.1 Absorption matrix

The interaction between the radiation and the atmosphere are described by the 4x4 absorption matrix $K$:

$$10 \quad K \quad = \quad k_a I + K_o \tag{1}$$

where $I$ is the identity matrix, $k_a$ is the scalar absorption coefficient and $K_o$ is a matrix with off-diagonal components:

$$K_o = \begin{bmatrix} 0 & q & u & v \\ q & 0 & v' & -u' \\ u & -v' & 0 & q' \\ v & u' & -q' & 0 \end{bmatrix} \tag{2}$$





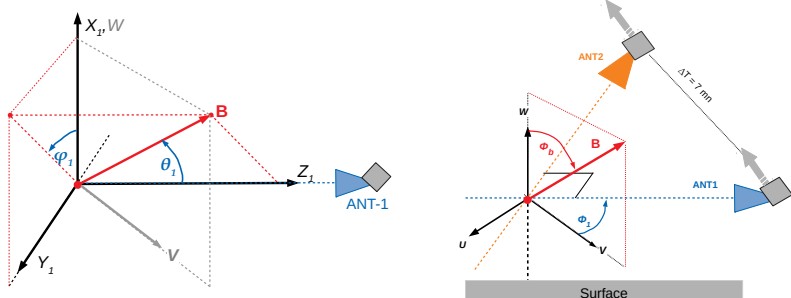

**Figure 4.** Left panel: Cartesian and spherical frames used for the radiative transfer calculation. The x- and z-axis are along the vertical axis (W) and the LOS, respectively. Right panel: Frame for describing the observation of the same air mass from the forward (ANT1) and aftward (ANT2) antennas. The background geomagnetic field ($B$) is at first approximation in the meridional plane.

The scalar absorption coefficient is computed using a line-by-line model and the Zeeman effect is only applied on the $O_2$ transition:

$$k_a(\nu,z) = \sum_{M,t} n_M(z) S_t(z) F(\nu,\nu_t,\Gamma_{M,z}) \; + \tag{3}$$
$$\frac{n_{O2}(z) S_x(z)}{2} \left( \sin^2(\theta) \sum_\pi \left[ s_\pi F(\nu,\nu_\pi,\Gamma_{O2,z}) \right] \; + \; \left( 1 + \cos^2(\theta) \right) \sum_{\sigma^+,\sigma^-} \left[ \frac{s_\sigma}{2} F(\nu,\nu_\sigma,\Gamma_{O2,z}) \right] \right)$$

where $\nu$ is the frequency, $z$ the altitude, $t$ denotes a spectroscopic transition of the species $M$ that is not affected by the geomagnetic field, $n_M$ ($n_{O2}$) is the number density of $M$ ($O_2$), $S_t$ is the line strength, $F$ is the Voigt function (Schreier et al., 2014; Larsson et al., 2014) and $\Gamma_{t,z}$ represents the parameters related to the linewidth (Appendix A). The angle $\theta$ is the inclination angle of the magnetic field with respect to the LOS (Fig. 4, left panel).

The frequencies $\nu_{\sigma,\pi}$ (Hz) are those of the Zeeman components (Fig. 3). They dependent on the magnetic field such as (Lars-
son et al., 2014):

$$\nu_{\sigma,\pi} = -\frac{\mu_b}{h_p} |B| g_s \beta_m \quad = 2.80209 \cdot 10^{10} |B| \beta_m \tag{4}$$

where $g_s = 2.002064$, $\mu_b$ is the Bohr magneton ($9.27401 \cdot 10^{-24}$ J T$^{-1}$), $h_p$ is the Planck constant ($6.62618 \cdot 10^{-34}$ m$^2$ kg s$^{-1}$) and

$$\beta_m = \left( \frac{J_u(J_u+1) + S(S+1) - N_u(N_u+1)}{2(J_u+1)J_u} m_u - \frac{J_l(J_l+1) + S(S+1) - N_l(N_l+1)}{2(J_l+1)J_l} m_l \right). \tag{5}$$

where the lower scripts $u$ and $l$ denote the upper and lower levels of the transition, respectively, $N$, $J$, $S$ and $m$ are quantum numbers associated with the angular momentum, the spin, the total momemtum $N+S$ and the projection of $J$ on the B axis.





**Table 4.** Zeeman transitions characteristics for the $O_2$ line at 773.84 GHz ($J : 4 \rightarrow 4$ and $S = 1$). The relative strengths are normalized such as $\sum s_\pi = \sum s_{\sigma+} = \sum s_{\sigma-} = 1$ (Tab. 3.1 in Landi Degl'Innocenti and Landolfi (2004)). The frequency shift factors $\beta_m$ are from Eq. 5.

| $m_u - m_l$ | $m_l$ | $s_{\pi,\sigma}$ | $\beta_m$ |
|:---:|:---:|:---:|:---:|
| $\pi$ | $\{-4 \ldots 4\}$ | $\frac{m_l^2}{60}$ | $\frac{9}{20} m_l$ |
| $\sigma^+$ | $\{-4 \ldots 3\}$ | $\frac{20 - m_l\,(m_l+1)}{120}$ | $\left(\frac{9}{20} m_l - \frac{1}{5}\right)$ |
| $\sigma^-$ | $\{-3 \ldots 4\}$ | $\frac{20 - m_l\,(m_l-1)}{120}$ | $\left(\frac{9}{20} m_l + \frac{1}{5}\right)$ |

The coefficients of $\boldsymbol{K}_o$ are derived from Landi Degl'Innocenti and Landolfi (2004) (Eq. 5.36):

$$q = \frac{\sin^2(\theta)\cos(2\phi)}{2} \left( \sum_\pi [s_\pi F(\nu, \nu_\pi, \Gamma_{x,z})] - \sum_{\sigma\pm} \left[\frac{s_\sigma}{2} F(\nu, \nu_\sigma, \Gamma_{O2,z})\right] \right)$$

$$u = \frac{\sin^2(\theta)\sin(2\phi)}{2} \left( \sum_\pi [s_\pi F(\nu, \nu_\pi, \Gamma_{x,z})] - \sum_{\sigma\pm} \left[\frac{s_\sigma}{2} F(\nu, \nu_\sigma, \Gamma_{O2,z})\right] \right) \qquad (6)$$

$$v = \cos(\theta) \left( \sum_{\sigma\pm} \pm \frac{s_{\sigma\pm}}{2} F(\nu, \nu_{\sigma\pm}, \Gamma_{O2,z}) \right)$$

The parameters $u'$, $v'$ and $q'$ are computed by replacing the term $F$ with $F'$, the dispersive part of the complex Voigt function (See Appendix A and (Schreier et al., 2014)).

## 3.2 Radiative transfer

The LOS is divided in narrow ranges of size $ds$ (typically 5 km long) in which the atmospheric parameters are considered constant. The change of the polarized radiance passing through an homogeneous range is derived from a matrix equation

which is similar to the scalar radiative transfer one used for a non-polarized radiation (Semel and López, 1999):

$$\boldsymbol{b}_a(s+ds) = \left(\boldsymbol{I} - \boldsymbol{\Lambda}(s, s+ds)\right) \cdot \boldsymbol{b}_p(s) + \boldsymbol{\Lambda}(s, s+ds) \cdot \boldsymbol{b}_a(s), \qquad (7)$$

where $\boldsymbol{b}_a(s)$ is the Stokes vector at the position $s$ on the LOS (the frequency dependence is omitted), "$\cdot$" is the matrix multiplication operator, $\boldsymbol{b}_p(s) = [P(s), 0, 0, 0]^T$ describes the non-polarized source function between $s$ and $s + ds$, $P(s)$ is the Planck function, and $\boldsymbol{\Lambda}(s, s+ds)$ is $4 \times 4$ evolution operator matrix defined as:

$$\boldsymbol{\Lambda}(s, s+ds) = \exp(-\boldsymbol{K}(s)\,ds). \qquad (8)$$

The integration over the LOS is performed by applying the scalar equation given by Urban et al. (2004) to Stokes parameters:

$$\boldsymbol{b}_a(\text{sat}) = \sum_{i=0}^{N-1} \boldsymbol{\Lambda}(i+1, \text{sat}) \cdot \left(\boldsymbol{I} - \boldsymbol{\Lambda}(0, i+1) \cdot \boldsymbol{\Lambda}(0, i+1)\right) \cdot \left(\boldsymbol{b}_p(i) - \boldsymbol{b}_p(i+1)\right) \qquad (9)$$

$$+ \left(\boldsymbol{I} - \boldsymbol{\Lambda}(0, \text{sat}) \cdot \boldsymbol{\Lambda}(0, \text{sat})\right) \cdot \boldsymbol{b}_p(N)$$

where $\boldsymbol{b}_a(\text{sat})$ is the Stokes vector representing the radiation state at the antenna position, $i$ is the index of the level at $s_i$ ($i = 0$

for the tangent point) and $N$ is the number of levels above the tangent point. The cosmic background radiation is neglected.





We use the relationship $\boldsymbol{\Lambda}(i,j) = \boldsymbol{\Lambda}(k,j) \cdot \boldsymbol{\Lambda}(i,k)$ with $i < k < j$ (the two matrices on the right-side of the equality do not commute).

## 4 Measurement and retrieval setting

### 4.1 Measured radiance

The measured radiance for antenna $a$ ($a =$1 or 2) at the elevation angle $\theta$ and the IF $\nu$ is:

$$y^a_{\theta,\nu} \;=\; \frac{1}{2} \left( \boldsymbol{R}^u_{\theta,\nu} \circledast \boldsymbol{y}^{a,u}(\nu_{LO} + \nu) \;+\; \boldsymbol{R}^l_{\theta,\nu} \circledast \boldsymbol{y}^{a,l}(\nu_{LO} - \nu) \right), \tag{10}$$

where $\boldsymbol{y}^{a,u}$ and $\boldsymbol{y}^{a,l}$ are the atmospheric specific intensities in the upper and lower sidebands around the local oscillator frequency $\nu_{LO}$, $\boldsymbol{R}_{\theta,\nu}$ represents the antenna and spectrometer functions and $\circledast$ is the convolution operator (Baron et al., 2018). A simple case with a constant upper and lower sideband ratios is considered. The Zeeman model is only used within

a bandwidth of 200 MHz encompassing the $O_2$ line (upper sideband). Outside this range, the non polarized radiative transfer model described in Baron et al. (2018) is used. In order to transform the Stokes vector (Eq. 9) to the specific intensity associated with the radiometer's polarization, we first rotate the vector from the atmospheric frame to the detector frame as:

$$\boldsymbol{b}_d \;=\; \boldsymbol{M_r}(\alpha_d) \cdot \boldsymbol{b}_a, \tag{11}$$

where $\boldsymbol{b}_d$ is the Stokes vector in the instrument frame and $\boldsymbol{M_r}(\alpha_d)$ is the Mueller matrix for a rotation $\alpha_d$:

$$\boldsymbol{M_r}(\alpha_d) = \begin{bmatrix} 1 & 0 & 0 & 0 \\ 0 & \cos(2\,\alpha_d) & \sin(2\,\alpha_d) & 0 \\ 0 & -\sin(2\,\alpha_d) & \cos(2\,\alpha_d) & 0 \\ 0 & 0 & 0 & 1 \end{bmatrix} \tag{12}$$

The specific intensity $y$ corresponding to the detector polarization is

$$y \;=\; \boldsymbol{b}_d[1] + m\,\boldsymbol{b}_d[n], \tag{13}$$

where $\boldsymbol{b}_d[n]$ is the $n^{th}$ component of the Stokes vector and $(m,n)$ is $(-1,1)$, $(1,1)$, $(-1,2)$, $(1,2)$, $(-1,3)$ and $(1,3)$ for horizontal, vertical, $+45°$, $-45°$, right and left circular polarizations, respectively.

Figure 6 shows simulated spectra of the $O_2$ line over the equator and at 80 °N when the satellite is moving toward north (ascending orbit branch). The tangent height is 100 km and the atmospheric conditions are representative of the northern hemisphere in wintertime (Baron et al., 2018). The magnetic field characteristics are zonal means inferred from a quiet solar day (Fig. 5). Spectra are shown for different radiometer's polarizations. Over the equator, $\boldsymbol{B}$ is along the meridional direction and clear differences are seen between the radiances measured with both antennas, except if the detector has a vertical

polarization. In that case, the radiometer detects only the $\sigma^{\pm}$ lines independently of the LOS orientation. The antenna-1 spectrum measured with a radiometer with a horizontal polarization is sensitive to the $\pi$ components which gives the visible double line shape. A receiver with a right-circular polarization measures mainly the $\sigma^+$ components since the antenna-1 is





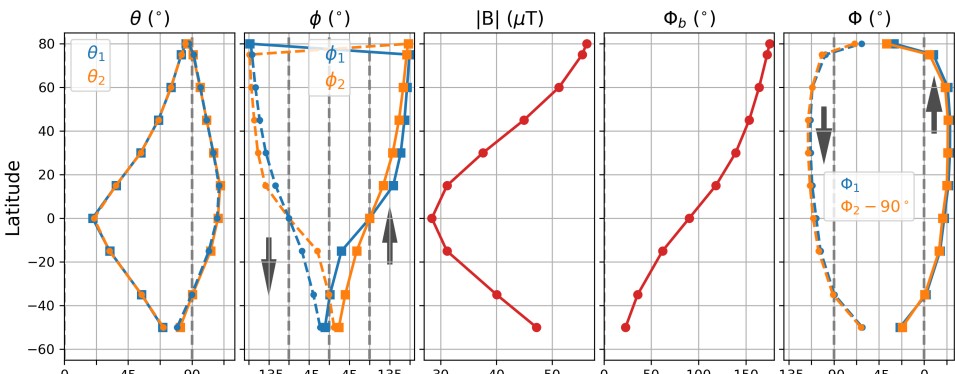

**Figure 5.** Magnetic field ($\theta_{1,2}$, $\phi_{1,2}$, $|\boldsymbol{B}|$ and $\Phi_b$) and LOSs ($\Phi_{1,2}$) parameters (Fig. 4) with respect to latitudes. The blue (yellow) lines are for ANT1 (ANT2) data. The circle-dashed (square-full) lines are data on the descending (ascending) orbit branch. The gray arrows in panels 2 and 5 indicate the direction of the satellite motion.

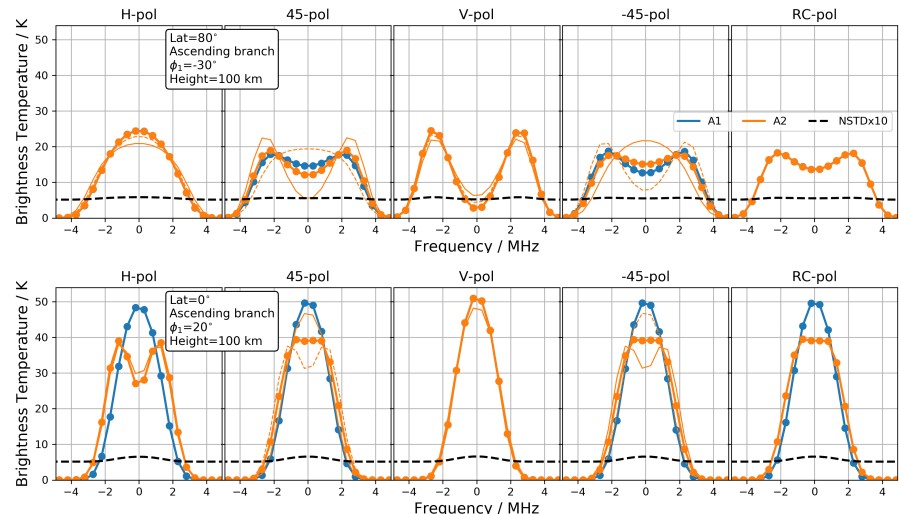

**Figure 6.** Upper panels: $O_2$ lines simulated for antenna-1 (blue) and antenna-2 (yellow) at $80°$N over the ascending orbit. Panels from left to right show the results for a detector with horizontal, $+45°$, vertical, $-45°$ and right-circular polarization. The dashed and full yellow thin lines are spectra calculated with an angular tilt of the antenna-2 detector of $-20°$ and $+20°$. The black dashed lines are the measurement noise STD$\times 10$ for antenna-1. Lower panels: Same as upper panels but for Equator.


nearly aligned with the magnetic field ($\theta_1 = 20°$ in Fig. 5). The spectrum looks like a single line with a frequency shift of $\Delta\nu \approx 420 \, (\beta_{m=-1} + \beta_{m=0}) = -21 \, \mathrm{kHz}$ (Eq. 4), equivalent to a LOS wind of $8 \, \mathrm{m \, s^{-1}}$.

Over the polar region, the spectra measured by both antennas are very similar since the vector $\boldsymbol{B}$ is almost vertical and perpendicular to both LOSs (Fig. 5). Only the Zeeman components $\pi$ are detected with the receiver with vertical polarization while the horizontally polarized one detects $\sigma^{\pm}$ components (Fig. 3).

## 4.2 Retrieval setting

The geomagnetic field may exhibit rapid temporal and spatial variations that can be as large as hundreds $\mathrm{nT}$ (Doumbia et al., 2007; Yee et al., 2017). Such variations will be difficult to take into account when processing the data and may lead to retrieval errors with the same magnitude as those induced by the measurement noise.

Such errors are mitigated by retrieving the three components of $\boldsymbol{B}$ simultaneously with other atmospheric parameters. It is done by using the scans of the same atmospheric column measured with the two antennas (Fig. 4). The measurement vector $\boldsymbol{y}$ is defined accordingly as:

$$\boldsymbol{y}^T = \begin{bmatrix} \boldsymbol{y}^{a1}, \boldsymbol{y}^{a2} \end{bmatrix} \tag{14}$$

The vector $\boldsymbol{x}$ describing the retrieved parameters contains the profiles of the chemical species having the most significant features in the MLT spectra, namely $O_2$, $H_2O$, $O_3$, $NO$ and $HDO$ (Fig. 2). It also includes the profiles of temperature T, line-of-sight winds (LW) and the three components of $\boldsymbol{B}$. It is defined as:

$$\begin{aligned} \boldsymbol{x}^T = [ \; & \boldsymbol{x}^{a_1}_{O2}, \cdots, \boldsymbol{x}^{a_1}_{T}, \boldsymbol{x}^{a_1}_{LW}, \\ & \boldsymbol{x}^{a_2}_{O2}, \cdots, \boldsymbol{x}^{a_2}_{T}, \boldsymbol{x}^{a_2}_{LW}, \\ & \boldsymbol{x}_{Bw}, \boldsymbol{x}_{Bu}, \boldsymbol{x}_{Bv} \; ], \end{aligned} \tag{15}$$

where the superscripts "a1" and "a2" denote that the parameters are associated with the antennas 1 and 2, respectively. The terms $\boldsymbol{x}_{Bw}$, $\boldsymbol{x}_{Bu}$, and $\boldsymbol{x}_{Bv}$ are the profiles of the vertical, zonal and meridional components of $\boldsymbol{B}$. The abundance and temperature profiles are retrieved for each antenna in order to account for differences between both scan locations. This is a similar approach as that used by Hagen et al. (2018) for the measurement of winds with the ground-based radiometer WIRA.

The retrieval error induced by the measurement noise is (Rodgers, 2000)

$$\epsilon_n^2 = \mathrm{diag}\left\{ \left( \boldsymbol{K}^T \boldsymbol{S}_y^{-1} \boldsymbol{K} + \boldsymbol{U}^{-1} \right)^{-1} \right\}, \tag{16}$$

where $\boldsymbol{K} = \frac{d\boldsymbol{y}}{d\boldsymbol{x}}$ is the Jacobian matrix of the retrieved parameters $\boldsymbol{x}$ and $\boldsymbol{U}$ is a diagonal matrix to ensure a stable inversion but with values large enough to allow us to neglect its effects in the altitude range where the retrievals are relevant (Baron et al., 2018). The matrix $\boldsymbol{S}_y$ is the diagonal covariance matrix associated with the measurement noise:

$$Sy_{i,i} = \frac{(T_{sys} + y_i)^2}{\delta\nu \, \delta t}, \tag{17}$$

and $Sy_{i,i}$ is the noise induced variance on the $i^{th}$ component of the measurement vector $\boldsymbol{y}$, $T_{sys}$ is the system temperature (Tab. 3), $\delta\nu$ the frequency resolution (0.5 MHz) and $\delta t$ the spectrum integration time (0.25 s).





The radiative transfer model computes the Jacobian $\boldsymbol{K}_B = \partial \boldsymbol{y}^{a_i}/\partial \boldsymbol{x}_B$ with respect to antenna-$i$ frame ($\{\boldsymbol{x_i}, \boldsymbol{y_i}, \boldsymbol{z_i}\}$ in left panel of Fig. 4). The matrix $\boldsymbol{K}_B$ is then computed in the atmospheric frame (Fig. 4):

$$\frac{\partial \boldsymbol{y}^{a_i}}{\partial \boldsymbol{B}_q} = \sum_{k=\{x_i, y_i, z_i\}} \frac{\partial \boldsymbol{y}^{a_i}}{\partial \boldsymbol{B}_k} \frac{\partial \boldsymbol{B}_k}{\partial \boldsymbol{B}_q}, \tag{18}$$

where $q = \{\boldsymbol{u}, \boldsymbol{v}, \boldsymbol{w}\}$ denote the atmospheric frame axes, and

$$\boldsymbol{B}_{x_i} = \boldsymbol{B}_w \tag{19}$$

$$\boldsymbol{B}_{y_i} = \cos(\Phi_i)\,\boldsymbol{B}_u + \sin(\Phi_i)\,\boldsymbol{B}_v$$

$$\boldsymbol{B}_{z_i} = -\sin(\Phi_i)\,\boldsymbol{B}_u + \cos(\Phi_i)\,\boldsymbol{B}_v$$

where $\Phi_i$ is the angle between the antenna-$i$ LOS and the meridional direction (left panel of Fig. 4).

## 5 Retrieval errors

Figure 7 shows the retrieval errors on the atmospheric density, temperature, LOS wind and the main chemical species at three latitudes (50°S, Equator, 80°N). For the instrumental setting, we considered a radiometer with a linear vertical polarization and the forward-looking antenna (antenna-1). The vertical resolution of the retrieved profiles is 2.5 km for the main parameters (temperature, LOS wind, $H_2O$ and $O_3$), 5 km for NO, and 20 km for the components of $\boldsymbol{B}$. Errors are computed for the same winter (DJF) climatology described in the previous section. The corresponding atmospheric state includes a stable polar vortex, and does not show any NO enhancement due to energetic particle precipitation. The results at the Equator and the southern hemisphere (SH) mid-latitudes (50°S) are for daytime conditions, while the northern hemisphere (NH) results at 80°N are representative of the polar night. We did not find significant differences between daytime and nighttime except for the relative error on $O_3$ retrieval, which is photo-dissociated between 60–80 km.

The results for the full band are compared with those computed for the inversion of a 200 MHz band containing only the $O_2$ line. The purpose is to isolate and characterize the contribution of the $H_2O$, $O_3$ and NO spectral lines to the retrieval of MLT parameters, in terms of altitude range and impact on the retrieval errors. Latitudinal differences are induced by the mean meridional circulation (from the summer pole to the winter pole). It is responsible for an increase (decrease) of NO ($H_2O$) in the winter hemisphere compared to the summer hemisphere, especially over the summer polar region. The largest sensitivity to NO is found in the upper part of the MLT. The precision is better than 10% above 95 km at 50°S and above 78 km over the winter polar region (NH in this study). A precision of 10% or better is achieved above 95 km at 50°S and 78 km in the winter polar region. The sensitivity to $H_2O$ decreases with increasing altitude, more sharply above 90 km. The precision is better than 1% up to 75 km in the SH and 65 km in the NH polar region.

The relative error on $O_3$ retrieval is $\sim 1\%$ around 60 km and strongly increases with increasing altitude and outside of the polar night, because of the daytime photo-dissociation of $O_3$.

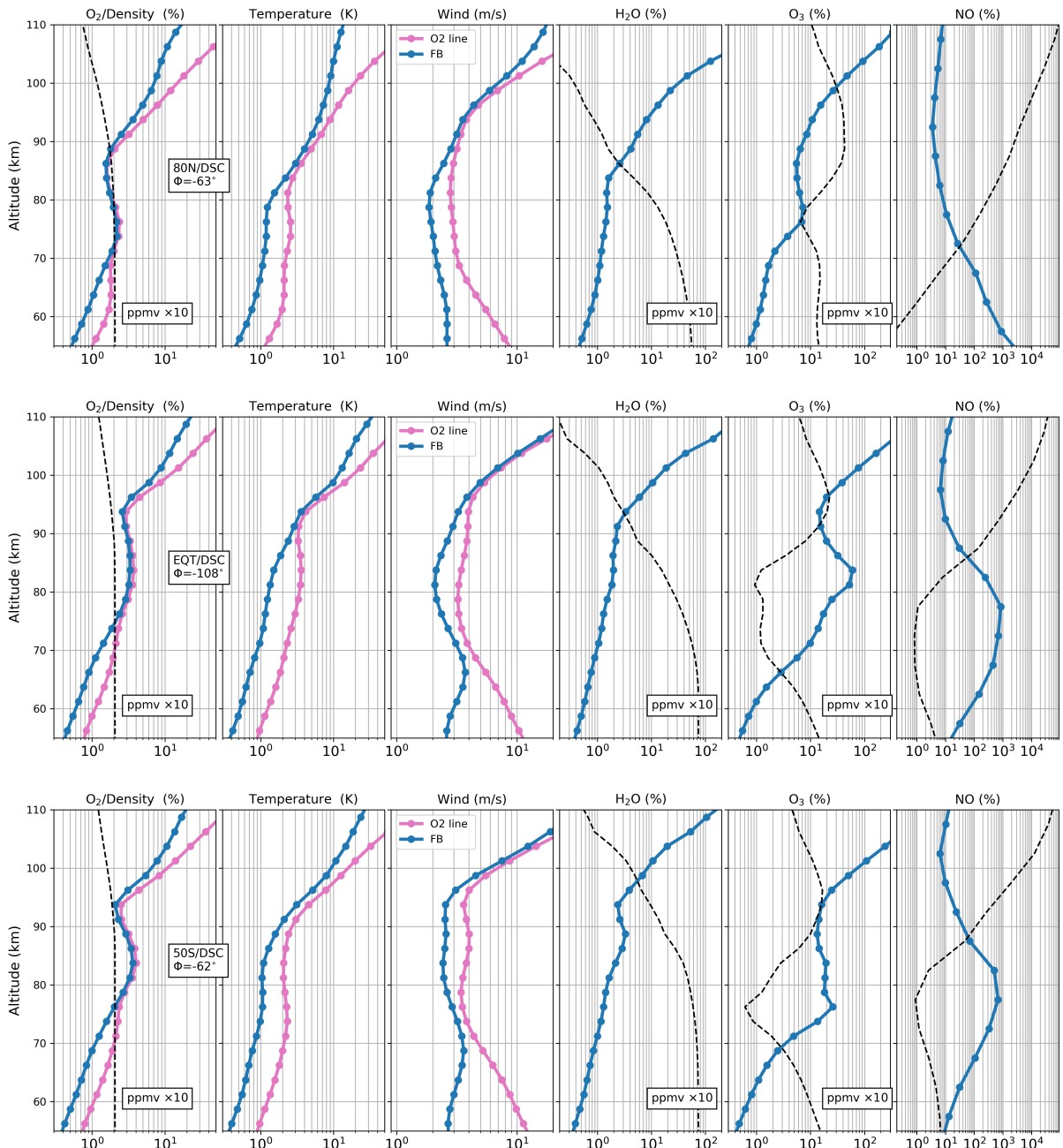

**Figure 7.** Errors on atmospheric density, temperature, LOS wind, $H_2O$, $O_3$ and NO profiles. Only errors induced by the measurement noise are shown (curves with filled symbols), for profiles retrieved from the antenna-1 signal using a radiometer with linear vertical polarization. The pink and blue curves show the results for the 200 MHz narrow band around the $O_2$ line and for the full bandwidth, respectively. Upper panel: results at 80°N (polar night). Middle and lower panels: results for daytime conditions at the Equator (middle) and 50°S (bottom). The vertical resolution for the retrievals is set to 2.5 km for all profiles, except for NO for which 5 km is used. The dashed black lines show the climatological profiles (ppmv).



## 5.1 Atmospheric density, temperature and LOS wind

The achieved precision of the atmospheric density (or $O_2$) profile is better than 5% up to about 95 km at all latitudes. Above 90 km, the signal intensity drops significantly and errors quickly increase, up to 20% at 110 km. Outside of the 70–90 km range, there are significant differences between the error profiles calculated for the full- and narrow-band inversions. This

shows that spectral lines from other molecular species also have an impact on the $O_2$ retrievals. This impact likely occurs through the temperature retrieval. For instance, over the winter polar region, the strong NO signal significantly improves the temperature retrievals, thus indirectly improves the $O_2$ abundance retrieval. Similarly, including $H_2O$ and $O_3$ lines leads to an improvement of the $O_2$ retrieval quality below 70 km.

For all latitudes, the temperature retrieval error is better than 5 K below 90 km and 30 K at 110 km. The $O_2$ line is the main

source of information on the temperature near 90 km.

The LOS wind, a key product for SMILES-2, is retrieved with a precision of 2–4 m s$^{-1}$ up to 90 km. Above this altitude, the retrieval errors strongly increase, up to 20 m s$^{-1}$ or more at 110 km. The $O_2$ line is the main source of information on the LOS wind above 70 km. Over the polar region and above 100 km, spectral lines of NO contribute significantly to the LOS wind retrievals.

Figures 8 and 9 show the achieved retrieval precisions for temperature and LOS wind, at altitudes between 80 and 110 km and for different polarization settings. Results are shown within the latitude range 50°S–80°N, for both antennas and for both the ascending and descending orbit branches. The results obtained with $\boldsymbol{B} = 0$ are also presented.

For atmospheric temperature and below 90 km, the Zeeman effect has a negligible impact on the retrieval errors. Differences can be seen only at high latitudes, where the decrease of the $H_2O$ abundance explains the larger impact of the $O_2$ line on the

retrieval. In terms of LOS wind retrieval, the Zeeman effect is negligible below 80 km. Above 90 km, the approximation $\boldsymbol{B} = 0$ leads to a significant underestimation of the retrieval errors, with differences of up to a factor of 2. This clearly shows that the retrieval errors depend on the radiometer polarization, the LOS orientation and on the characteristics of the magnetic field.

Best overall precision is found for a radiometer with a linear vertical polarization. For instance, at NH high latitudes, the LOS wind retrieval error at 99 km is 6 m s$^{-1}$ using a linear vertical polarization, but degrades to about 10 m s$^{-1}$ for other

polarization settings. Furthermore, using the linear vertical polarization yields homogeneous results for different observation geometries: we could not find significant differences between ascending and descending orbits or between the two antennas.

## 5.2 Geomagnetic field

Figure 10 shows the retrieval errors on the three components of $\boldsymbol{B}$ at 85 km and 105 km (vertical resolution of 20 km). The results strongly depend on the radiometer's polarization. Best performance is achieved with a ±45° linear polarization. Errors

are clearly smaller when the retrieved component is aligned with the background magnetic field: the error on $B_v$ is smallest at the equator where $\boldsymbol{B}$ is horizontal and in the meridional plane, and the error on $B_w$ minimizes at high latitudes where $\boldsymbol{B}$ is nearly vertical. The best sensitivity is found at 85 km where the precision is better than 400 nT for all components and at all latitudes, except for the zonal component ($B_u$) in the tropics. At high latitudes, errors are between 50–100 nT for the vertical





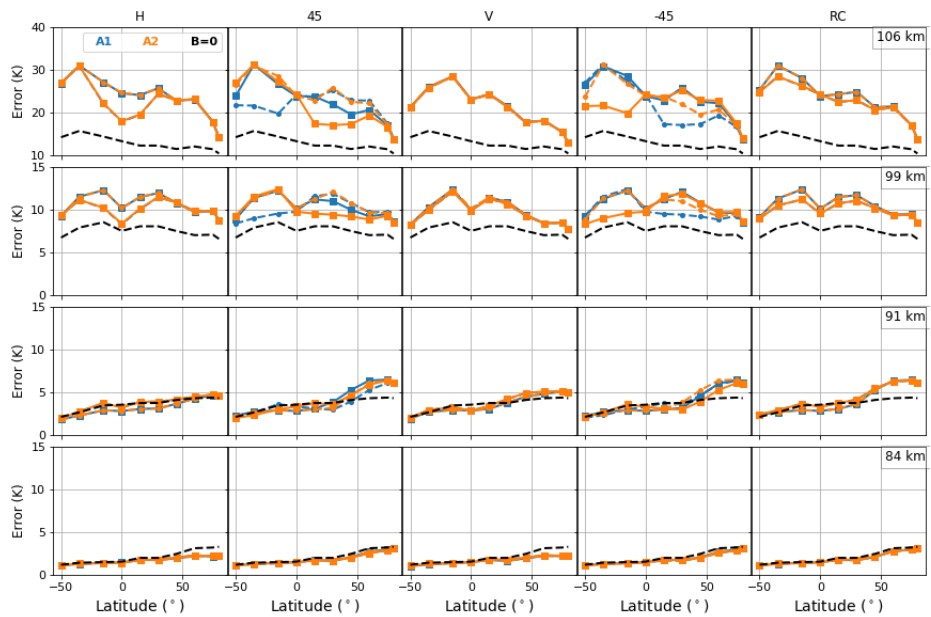

**Figure 8.** Retrieval errors on the temperature profile induced by the measurement noise, for a radiometer with horizontal (H), 45° (45), vertical (V) and -45° (-45) linear polarizations, and right circular (RC) polarization. Colored dashed (full) lines: results for the descending (ascending) orbit branch. The blue (yellow) lines show the results for antenna-1 (antenna-2). The black dashed lines show the errors if the Zeeman effect is not considered. Vertical resolution of the retrieved profiles is 2.5 km.

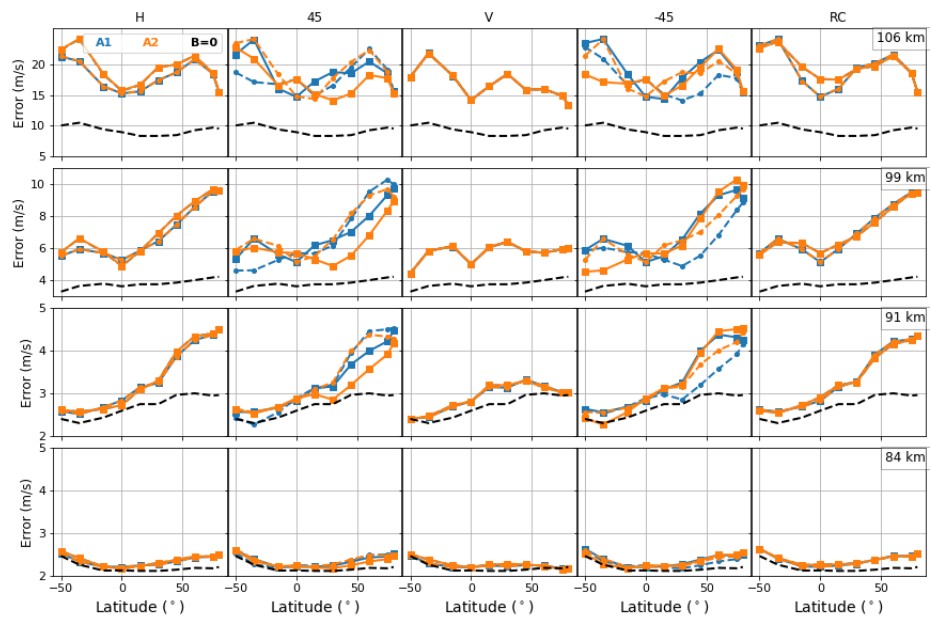

**Figure 9.** Same as Fig. 8 but for line-of-sight wind retrievals.



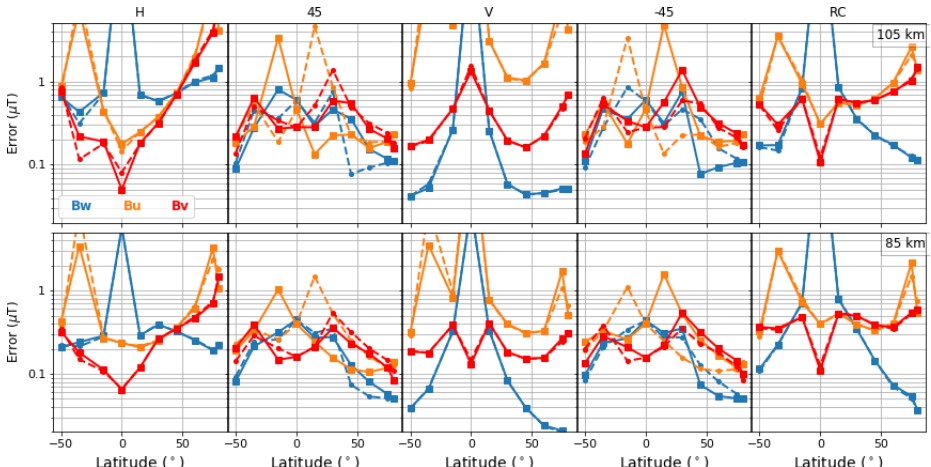

**Figure 10.** Same as Fig. 8 but for the geomagnetic fields components and altitudes between 85 and 105 km. The retrieval vertical resolution is 20 km.

component ($B_w$) and 100–200 nT for the horizontal ones ($B_u$ and $B_v$). At 105 km, errors increase, for example to 80–500 nT outside the tropics.

Contrary to the results shown in Sect. 5.1 where it was the optimal configuration, the linear vertical polarization yields a worse retrieval performance for $\boldsymbol{B}$. In this case, the retrieval errors on $\boldsymbol{B}_w$ and $\boldsymbol{B}_u$ over the tropics are much larger than those

found with a slant polarization. Only the meridional component ($\boldsymbol{B}_v$) can still be retrieved with a reasonable precision of 100–400 nT. At mid- and high latitudes, best precision is found for $\boldsymbol{B}_w$ (30–50 nT at 85 km and 50–70 nT at 105 km). At 85 km, the error on $\boldsymbol{B}_v$ and $\boldsymbol{B}_u$ are between 200–300 nT and 300–2000 nT, respectively. Large errors on $\boldsymbol{B}_u$ are found at 40°S and 70°N where the LOS is aligned with the $U$ and $V$ axes ($\Phi_1$ or $\Phi_2 = 0$, Fig. 5).

Our results show that the sensitivity of the SMILES-2 instrument is high enough to potentially measure the electrojet induced

variations of $\boldsymbol{B}$ at high latitudes even under quiet sun conditions, provided that the data are properly averaged. Yee et al. (2017) used the Zeeman effect on the AURA/MLS $O_2$ line to derive variations of 100–200 nT on the intensity of $\boldsymbol{B}$. During solar storms, the amplitude of the perturbations in the auroral regions could be considerably larger (several hundreds of nT) (Yee et al., 2017; Yamazaki and Maute, 2017) and could be detected with single measurements along the vertical and at least one horizontal component of $\boldsymbol{B}$. Hence, SMILES-2 could allow us to infer information on the 3D variations of the auroral electrojet.

Perturbations of the geomagnetic field near the equator (30 nT and 80 nT for the surface vertical and horizontal components of $\boldsymbol{B}$) are much smaller than the retrieval precision (Doumbia et al., 2007). Therefore, extracting interesting information on the equatorial jet will be more challenging and a receiver with a slant polarization could be necessary.





## 6    Conclusions

This analysis demonstrates the potential of SMILES-2 for the measurement of the temperature, atmospheric density and LOS wind in the MLT (60–110 km). The retrieval precision was assessed, focusing on the SMILES-2 band at 760 GHz, the most suitable for such measurements. Special care was taken to properly include the Zeeman effect on the $O_2$ line. Our results
showed that neglecting it could lead to underestimating the retrieval errors by a factor of up to 2 above 90 km. Because the $O_2$ line is polarized, the radiometer's polarization configuration had to be investigated. We found that the optimal configuration was vertical linear. The line-of-sight wind is retrieved with a precision of 2–5 $\mathrm{m\,s^{-1}}$ up to 90 km (30 $\mathrm{m\,s^{-1}}$ at 110 km) and a vertical resolution of 2.5 km. Temperature and atmospheric density are retrieved with a precision better than 5 K (30 K) and 7% (20%) up to 90 km (110 km), respectively. The achieved precision of the wind measurements, a key product for SMILES-2, is
comparable to the requirements for the new ICON mission (Englert et al., 2017). However, unlike optical sensors, SMILES-2 can acquire high-precision measurements during day and night, and at all latitudes, even during auroral events. The low noise level achieved by the 4-K super-cooled radiometers is essential to achieve good performance above 90 km, where sensitivity becomes critical due to significantly weaker signals.

The retrieval of the geomagnetic field using the $O_2$ line was also discussed. We showed that valuable information on the
horizontal and vertical components of $\boldsymbol{B}$ could be determined directly near the E-region auroral electrojets. Yee et al. (2017) highlighted the need for such observations since, currently, only measurements from the ground or from low-orbit satellites near 400 km are available. Yee et al. proposed a cubeSat constellation, with the purpose of measuring the $O_2$ line at 119 GHz to produce high spatial and temporal observations of $\boldsymbol{B}$ perturbations. It is worth mentioning that this methodology, based on $O_2$ spectral lines, has also been proposed to measure the Martian residual magnetic field (Larsson et al., 2013). Further analyses
should be conducted, to characterize more precisely the potential of SMILES-2 for the study of the 3D ionospheric electrojets.

The final instrumental setup is still under discussion. In terms of possible instrumental developments, the spectral bandwidth of the 763-GHz band might be reduced in the definitive configuration of SMILES-2. Narrowing the bandwidth by a factor of 2 (while ensuring a correct adjustment of the LO frequency) would cause minimal degradation of the measurement performance, limited to altitudes below about 40 km.

Future work to improve MLT retrievals will include the two other SMILES-2 bands. Indeed, the atomic oxygen line at 2 THz contains temperature and wind information above 100 km. This line can help us to improve the wind retrieval precision to 10 $\mathrm{m\,s^{-1}}$ at 110 km (Baron et al., 2019). Below about 70 km (90 km), a strong signal form $O_3$ will be measured during daytime (nighttime) in the 638-GHz band. Furthermore, new parameters for the Zeeman model became recently available (Larsson et al., 2019). Applying the updated parameters should induce a change of the $O_2$ and O line intensities, of up to a few percent.
The Zeeman effect on other spectral lines: OH, NO and ClO, should also be studied.

## Appendix A:  Spectroscopic parameters





**Table A1.** Parameters of $O_2$ lines in the ground electronic and vibrational levels between 100 and 1000 GHz. Values are taken from the HITRAN-2008.

| Frequency | Strength @ 296K | $E_L$ | N | J |
|---|---|---|---|---|
| MHz | $cm^{-1}$ molecule$^{-1}$ cm$^2$ | cm$^{-1}$ | lower,upper | lower,upper |
| 118750.3408 | 9.956e-26 | 0.0000 | 1,1 | 0,1 |
| 368498.3839 | 2.213e-26 | 3.9611 | 1,3 | 1,2 |
| 424763.1626 | 2.414e-25 | 2.0843 | 1,3 | 2,2 |
| 487249.4142 | 1.031e-25 | 2.0843 | 1,3 | 2,3 |
| 715393.1236 | 6.246e-26 | 18.3372 | 3,5 | 3,4 |
| 773839.7019 | 3.943e-25 | 16.3876 | 3,5 | 4,4 |
| 834145.7729 | 1.359e-25 | 16.3876 | 3,5 | 4,5 |

The spectroscopic parameters are taken from the HITRAN database (Rothman et al., 2009). The line strength at the temperature $T$ is:

$$S(T) = \frac{C_H}{r_{\text{iso}}} S_H(T_0) \frac{e^{-C_E E_L/k_b T}}{e^{-C_E E_L/k_b T_0}} \left( \frac{1 - e^{-C_E \bar{\nu}_0/k_b T}}{1 - e^{-C_E \bar{\nu}_0/k_b T_0}} \right) \frac{Q(T_0)}{Q(T)} \qquad (\text{Hz m}^2 \text{ molecule}^{-1}) \qquad (A1)$$

where $k_b = 1.380662 \times 10^{-23}$ J K$^{-1}$ is the Bolzmann constant, $\bar{\nu}_0$ (cm$^{-1}$) is the transition wavenumber, $S_H(T_0)$ is the HI-
TRAN line strength (cm$^{-1}$ cm$^2$ molecule$^{-1}$), $T_0 = 296$ K, $E_L$ (cm$^{-1}$) is the lowest energy of the transition. The partition function $Q$ is calculated from tabulated values between 120 and 500 K, a range that encompasses the temperatures found between 50 and 130 km ($Q(296) = 215.77$). The constants $C_E = 10^2 h_p c$ and $C_H = 10^{-2} c$ allow the conversion of the HI-TRAN units to the International System (SI) ones. The isotopic ratio $r_{\text{iso}}$ is taken away from $S_H$ and added to the density profile instead. The table A1 shows parameters of the main $O_2$ millimeter lines.

Above the altitude of about 70 km, the real part of the Voigt function $F$ (Eq. 3) is close to the Gauss function that describes lines broadened by random molecular velocities (Doppler broadening):

$$F(\nu) = \frac{1}{\Delta\nu_d} \left( \frac{\ln 2}{\pi} \right)^{\frac{1}{2}} e^{-\ln 2 \left( \frac{\nu - \nu_0}{\Delta\nu_d} \right)^2} \qquad (\text{Hz}^{-1}) \qquad (A2)$$

with

$$\Delta\nu_d = \frac{\nu_0}{c} \left( \frac{2\ln 2\, R T}{M} \right)^{\frac{1}{2}} \qquad (\text{Hz}), \qquad (A3)$$

and $\Delta\nu_d$ is the Doppler broadening parameter, i.e., the Half Width at Half Maximum (HWHM) of $F$, $\nu_0$ is the frequency of the transition, $c = 2.997924 \times 10^8$ m s$^{-1}$ is the speed of light in vacuum, $R = 8.31446$ J K$^{-1}$ mol$^{-1}$ the gas constant and $M$ the molar mass (0.031980 kg mol$^{-1}$ for $O_2$). At 80 km, $\Delta\nu_d$ is about 0.6–0.7 MHz for the $O_2$ line at 773 GHz, while the pressure broadening HWHM is only 0.01–0.02 MHz.

The dispersion profile used for the calculation of the coefficient $q'$, $u'$ and $v'$ (Eq. 2) is given by:

$$F'(\nu) = \sqrt{2} F(\nu) \text{erfi}(v) \qquad (\text{Hz}^{-1}), \qquad (A4)$$





with $v = \ln 2 \left( \frac{\nu - \nu_0}{\Delta \nu_d} \right)$ and $\mathrm{erfi}(v) = \sqrt{2/\pi} \int_0^v \exp(t^2) \, dt$ is the imaginary error function (Eq. 5.54 in Landi Degl'Innocenti and Landolfi (2004)).

**Appendix B: Matrix exponential**

5  The computation of the matrix exponential in Eq. 1 is the performance bottleneck in our implementation of the radiative transfer solver if we use a general algorithm. A significantly faster algorithm has been implemented using the symmetry in $\boldsymbol{K_o}$ (Eq. 2). The evolution operator $\boldsymbol{\Lambda}$ (Eq. 8) is written as $\exp(-k_a \, ds) \exp(\tilde{\boldsymbol{K}}_o)$ with $k_a$ the scalar absorption coefficient (Eq. 3) and $\tilde{\boldsymbol{K}}_o = -\boldsymbol{K}_o \, ds$. The Cayley-Hamilton theorem is used to compute $\exp(\tilde{\boldsymbol{K}}_o)$:

$$\exp(\tilde{\boldsymbol{K}}_o) = \sum_{k=0}^{3} \kappa_k \, \tilde{\boldsymbol{K}}_o^k, \tag{B1}$$

10  where $\tilde{\boldsymbol{K}}_o^0$ is the identity matrix. The coefficient $\kappa_k$ are derived using the 4 eigenvalues of $\tilde{\boldsymbol{K}}_o$:

$$\begin{cases} \exp^{\lambda_1} &= \kappa_0 + \kappa_1 \lambda_1 + \kappa_2 \lambda_1^2 + \kappa_3 \lambda_1^3 \\ \exp^{-\lambda_1} &= \kappa_0 - \kappa_1 \lambda_1 + \kappa_2 \lambda_1^2 - \kappa_3 \lambda_1^3 \\ \exp^{j \lambda_2} &= \kappa_0 + j \kappa_1 \lambda_2 - \kappa_2 \lambda_2^2 - j \kappa_3 \lambda_2^3 \\ \exp^{-j \lambda_2} &= \kappa_0 - j \kappa_1 \lambda_2 - \kappa_2 \lambda_2^2 + j \kappa_3 \lambda_2^3 \end{cases}$$

where $\lambda_{1,2}$ are positive real-valued numbers that determine the 4 eigenvalues $\pm \lambda_1$ and $\pm j \lambda_2$ of $\tilde{\boldsymbol{K}}_o$. This gives:

$$\begin{aligned} \kappa_0 &= \frac{\lambda_2^2 \cosh(\lambda_1) + \lambda_1^2 \cos(\lambda_2)}{\lambda_1^2 + \lambda_2^2} \\ \kappa_1 &= \frac{\lambda_2^2 \sinh(\lambda_1)/\lambda_1 + \lambda_1^2 \sin(\lambda_2)/\lambda_2}{\lambda_1^2 + \lambda_2^2} \\ \kappa_2 &= \frac{\cosh(\lambda_1) - \cos(\lambda_2)}{\lambda_1^2 + \lambda_2^2} \\ \kappa_3 &= \frac{\sinh(\lambda_1)/\lambda_1 - \sin(\lambda_2)/\lambda_2}{\lambda_1^2 + \lambda_2^2} \end{aligned} \tag{B2}$$

The eigenvalue parameters are $\lambda_1 = -ds \sqrt{(A+B)/2}$ and $\lambda_2 = -ds \sqrt{(A-B)/2}$, where

$$\begin{aligned} A = [ \quad & 8 \left( q q' v v' + u u' q q' + u u' v v' \right) + q^4 + u^4 + v^4 + q'^4 + u'^4 + v'^4 \\ + 2 ( \quad & q^2 (u^2 - u'^2 + v^2 - v'^2 + q'^2) + \\ & u^2 (u'^2 + v^2 - v'^2 - q'^2) + \\ & v^2 (-u'^2 + v'^2 - q'^2) + \\ & v'^2 (u'^2 + q'^2) + \\ & u'^2 q'^2 ) \, ]^{\frac{1}{2}}. \end{aligned}$$





and $B = q^2 - q'^2 + u^2 - u'^2 + v^2 - v'^2$.

*Competing interests.* The authors declare that they have no conflict of interest

*Acknowledgements.* The SMILES-2 study is supported by the strategic development research fund from the Institute of Space and Astronautical Science (ISAS)/JAXA. H.L. acknowledges support by JSPS KAKENHI grants 18H01270, 18H04446, and 17KK0095. P.B. would
5 like to thank Franz Schreier (German Aerospace Center, DLR) for providing the python implementation of the complex Voigt function used in GARLIC (see reference given for Eq. 3).





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
