# Peer review of "Potential for the measurement of MLT wind, temperature, density and geomagnetic field with Superconducting Submillimeter-Wave Limb-Emission Sounder-2 (SMILES-2)"

_Atmospheric Measurement Techniques, 2019_

## Referee Comment (RC1) · Hugh C. Pumphrey (Referee) · 7 Aug 2019

**General remarks**

This paper assesses the ability of a proposed sub-mm-wave limb sounding instrument to measure some dynamic properties of the mesosphere and lower thermosphere (MLT). No such instrument with the ability to measure both vector components of the wind has been flown before. Furthermore, at least some of the spectral regions which

the new instrument will use (including the spectral region described in this paper) have not been used by previous instruments. The paper therefore represents an advance in understanding and should be published.

The standard of written English in the paper is very good. Only a few mis-placed or missing "the"s and "a"s hint that it was not written by a native speaker. I note a few such corrections below. I do not claim to have found all of them, but in any case they should not prevent publication of the paper. The figures are mostly of an adequate quality for publication; I make a few suggestions for improvements below.

I am not a fan of sentences of the form "British (French) people prefer beer (wine)". There are several such sentences in the paper, e.g Page 1 lines 11–12, Page 13, line 22, and the caption of figure 8. I can not find any guidance in AMT's instructions as to whether such sentences are recommended or not, but I seem to recall dealing with another journal recently which told authors to avoid this construction. I do not insist that these sentences be re-written, but I suggest to the authors that they consider doing so. To write "British people prefer beer, while the French prefer wine" is a little more wordy, but far less confusing to the reader.

The one scientific/technical issue I have with the paper is that the explanation of how the quantities of temperature, pressure, density and height are connected seems inadequate. These quantites are linked via the hydrostatic equation and the ideal gas equation. Limb sounder data is typically used to estimate quantities as a function of some vertical co-ordinate.

- If that co-ordinate is pressure (as with MLS), then temperature is determined as a function of pressure, and the geopotential height of a single pressure surface is also estimated. The geopotential height of any other pressure surface is then obtained using the hydrostatic equation.

- If the vertical co-ordinate is geometric or geopotential height, the limb sounder

measures pressure and temperature as a function of height.

In both cases, density can be obtained from the pressure and temperature using the ideal gas equation. One could choose to use density instead of pressure as a vertical co-ordinate. Or, with height as a vertical co-ordinate, one could estimate the density directly from the measurements and obtain the pressure from the ideal gas equation. This appears to be the approach taken by the authors of this paper; I think they should explain it in a little more detail and state why they chose it over other possible approaches. They should in particular explain if, and if so, how, the hydrostatic equation is used to constrain their estimates.

**Specific comments**

- Page 2 line 31: MLS only measures one of the two vector components of the wind: it is perhaps worth reminding the reader here that the proposed instrument (with its two antennae) can measure both components.

- Page 3 lines 5-10 and table 1: This section should perhaps also reference Wang et al. (https://doi.org/10.5194/amt-2019-212), who describe the TALIS mission proposed by China. (TALIS, if built, will be very similar to Aura MLS.)

- Page 7 figure 3: The caption gives a magnetic field in Gauss, while Figure 10 shows magnetic fields in Tesla. The paper should present all magnetic fields in consistent units (preferably in Tesla as it is the SI unit.)

- Page 13, Line 10 and Page 14, Figure 7. The text states that the left panel shows "atmospheric density", while the label on the figure itself says "$O_2$/Density (%)". The figure appears to show neither of these things — rather, it shows the mixing ratio of $O_2$. It is quite confusing that this figure combines errors in % of the

quantity measured with profiles of the quantity itself. For $O_2$, the label "ppmv $\times$ 10" is wrong as the quantity shown is the mixing ratio itself. For NO, it is unclear how the scale should be interpreted for the climatological profile — maybe that panel is missing a label along the lines of "ppmv $\times 10^x$" for some $x$.

• Page 17, figure 10: many of the curves go off the top of the figures. It would be preferable to add an extra log decade at the top of each panel, while making the entire figure taller so that the same level of detail is visible as at present.

**Technical corrections**

• page 4 line 15: "Tables 2" should be "Table 2"

• Page 4 line 17: "with 7 min" should be "with a 7 min"

• Page 5 Line 3: "oxyde" should be "oxide"

• Page 5 line 8: "under the local" should be "under local"

• Page 6 Line 4: "The molecular" should just be "Molecular". Also, "so-call" should be "so-called", but actually I would remove it entirely — to a native English speaker "so-called" implies that people call it that, but that they are wrong to do so.

• Page 8 line 9: "They dependent" should be "they are dependent". Also, remove "such as" from this sentence.

• Page 10 lines 20 and 23: First mentions of figures in the text should be in numerical order. Here, Figure 6 is mentioned before figure 5.

- Page 12 Line 20: "a1" and "a2" should be in italics and should not have "" round them ($a1$ and $a2$), in order to match their appearance in the equations.

- Page 15 line 5. "likely" should be "probably". Despite ending in "-ly", likely is an adjective and is synonymous with "probable". It is not an adverb synonymous with "probably". You could re-word the sentence to read "It is likely that this impact occurs . . . ". (Americans never get this right, and I recognise that I am fighting a losing battle with them.)

---

## Referee Comment (RC2) · Anonymous Referee #3 · 12 Nov 2019

The study of Baron et al. is of high value for the readers of AMT. They well describe the details of the retrieval of a planned satellite mission for limb sounding the MLT region. Their simulation study yields the uncertainties which can be expected for the retrieved atmospheric parameters. A new point is the inclusion of the geomagnetic field in the retrieval process. It is beyond my expertise to verify the equations of the article. However, the article is carefully written and well explained. I only found some minor corrections. Thus, I recommend a publication of this article in AMT.

Minor corrections:

Table 1 W, T and NLC denote wind, temperature and . . .

page 5, line 12 molecular instead of moelcular

page 7, line 5 what does wrt mean?

page 8, line 9 They are dependent on . . .

page 10, line 9 . . . lower sideband ratio is . . ..

---

## Author Comment (AC3) · 15 Nov 2019

see supplement file

Please also note the supplement to this comment:
https://www.atmos-meas-tech-discuss.net/amt-2019-217/amt-2019-217-AC3-supplement.pdf

———————————————————

---

## Author Response (AR1)

**Reviewer 1**

We would to thank Hugh Pumphrey for his valuable comments on our manuscript. We believe they are all included in the revised manuscript uploaded as a supplement file. The changes are highlighted in red.

Herebelow we answer the comments point by point. (Reviewer' comments are indicated with [REV] and our answers with [ANS]).

[REV] To write "British people prefer beer, while the French prefer wine" is a little more wordy, but far less confusing to the reader.

[ANS] We agree and changed the sentences using this construction (Page 1 lines 11–12, Page 13, line 22, and the caption of figure 8).

[REV] The one scientific/technical issue I have with the paper is that the explanation of how the quantities of temperature, pressure, density and height are connected seems inadequate. These quantites are linked via the hydrostatic equation and the ideal gas equation. Limb sounder data is typically used to estimate quantities as a function of some vertical co-ordinate.
    • If that coordinate is pressure (as with MLS), then temperature is determined as a function of pressure, and the geopotential height of a single pressure surface is also estimated. The geopotential height of any other pressure surface is then obtained using the hydrostatic equation.
    • If the vertical co-ordinate is geometric or geopotential height, the limb sounder measures pressure and temperature as a function of height.

[ANS] We described this point in Sect. 2.3 too briefly. A specific session has been added to improve the discussion including details given by the reviewer. In short, we consider that the LOS tangent altitudes above 60 km are known when the spectra are inverted. They are inferred from the satellite attitude and by the extrapolation of LOS altitudes retrieved below 70 km from the measured radiances (e.g., using the MLS method). Based on our experience with JEM/SMILES, tangent heights derived from GPS will be better than 100 m (Ochiai et al., 2013).
The hydrostatic equilibrium equation is not used above 60 km to calculate the LOS tangent heights.

Ochiai, S. et al..: Receiver Performance of the Superconducting Submillimeter-Wave Limb-Emission Sounder (SMILES) on the International Space Station, IEEE Transactions on Geoscience and Remote Sensing, 51, 3791–3802, https://doi.org/10.1109/TGRS.2012.2227758, 2013.

**Specific comments**

[REV] Page 2 line 31: MLS only measures one of the two vector components of the wind: it is perhaps worth reminding the reader here that the proposed instrument (with its two antennae) can measure both components.

[ANS] We agree and the following sentence is added in the revised manuscript.
"… and MLS, which is equipped with a single antenna, can only measure one component of the wind vector (it was not designed for wind measurement)."

As for the 2 antennas on SMILES2, the information is given in P3L16.

[REV] Page 3 lines 5-10 and table 1: This section should perhaps also reference Wang et al. (https://doi.org/10.5194/amt-2019-212), who describe the TALIS mission proposed by China. (TALIS, if built, will be very similar to Aura MLS.)

[ANS] We agree and the following sentence has been added P3L9:
"TALIS, a limb sounder using similar spectral bands as Aura MLS, is being studied in China~\citep{Wang2019}."

[REV] Page 7 figure 3: The caption gives a magnetic field in Gauss, while Figure 10 shows magnetic fields in Tesla. The paper should present all magnetic fields in consistent units (preferably in Tesla as it is the SI unit.)

[ANS] It is corrected in the new manuscript.

[REV] Page 13, Line 10 and Page 14, Figure 7. The text states that the left panel shows "atmospheric density", while the label on the figure itself says "O2/Density (%)". The figure appears to show neither of these things — rather, it shows the mixing ratio of O2. It is quite confusing that this figure combines errors in % of the paper quantity measured with profiles of the quantity itself. For O2, the label "ppmv × 10" is wrong as the quantity shown is the mixing ratio itself. For NO, it is unclear how the scale should be interpreted for the climatological profile — maybe that panel is missing a label along the lines of "ppmv ×10x " for some x.

[ANS] New plots have been created (Figures 7 and 8) in which the atmospheric profiles and retrieval errors are shown separately. To avoid confusion, only atmospheric density is shown in the figures and indicated in the legend.

[REV] For O2, the label "ppmv × 10" is wrong as the quantity shown is the mixing ratio itself

[ANS] We agree, thanks for pointing out this mistake. In the new figure (Fig. 7 in the revised version), atmospheric profiles are shown without using scaling such as "unit x 10".

[REV] Page 17, figure 10: many of the curves go off the top of the figures. It would be preferable to add an extra log decade at the top of each panel, while making the entire figure taller so that the same level of detail is visible as at present.

[ANS] The abscissa ranges in Fig 7 (Fig. 8 in the revised version) has been increased as demanded by the reviewer.

**Technical corrections**

[REV] Page 4 line 15: "Tables 2" should be "Table 2"
[ANS] Done

[REV] Page 4 line 17: "with 7 min" should be "with a 7 min"

[ANS] Done

[REV] Page 5 Line 3: "oxyde" should be "oxide"
[ANS] Done

[REV] Page 5 line 8: "under the local" should be "under local"
[ANS] Done

[REV] Page 6 Line 4: "The molecular" should just be "Molecular". Also, "so-call" should be "so-called", but actually I would remove it entirely — to a native English speaker "so-called" implies that people call it that, but that they are wrong to do so.
[ANS] "The molecular" is corrected and "so-call" removed in the revised version.

[REV] Page 8 line 9: "They dependent" should be "they are dependent". Also, remove "such as" from this sentence.
[ANS] Done

[REV] Page 10 lines 20 and 23: First mentions of figures in the text should be in numerical order. Here, Figure 6 is mentioned before figure 5.
[ANS] The order of the figures is inverted in the revised version.

[REV] Page 12 Line 20: "a1" and "a2" should be in italics and should not have "" round them (a1 and a2), in order to match their appearance in the equations.
[ANS] Done. Also the sentence "where … " is moved to the previous equation (Eq 14).

[REV] Page 15 line 5. "likely" should be "probably". Despite ending in "-ly", likely is an adjective and is synonymous with "probable". It is not an adverb synonymous with "probably". You could re-word the sentence to read "It is likely that this impact occurs . . . ". (Americans never get this right, and I recognise that I am fighting a losing battle with them.)

[ANS] "probably" is used in the revised version.
Thanks for pointing this language issue. I will try to remember it.

**Reviewer 3**

We would like to thank reviewer 3 for his comments that helped us to improve our manuscript. Our replies are given here-below, and the manuscript has been modified accordingly in a supplement file. The manuscript corrections are highlighted in blue in the revised manuscript.

Herebelow we answer the comments point by point. (Reviewer' comments are indicated with [REV] and our answers with [ANS]).

[REV] Table 1 W, T and NLC denote wind, temperature and . . .

[ANS] Done. The sentence in the revised version is: "W, T and NLC denote wind, temperature and noctilucent cloud."

[REV] Page 5, line 12 molecular instead of moelcular

[ANS] Done

[REV] Page 7, line 5 what does wrt mean?

[ANS] It is an abbreviation for "with respect to". It is replaced by the full form in the revised manuscript.

[REV] Page 8, line 9 They are dependent on . . .

[ANS] Done

[REV] Page 10, line 9 . . . lower sideband ratio is . . ..

[ANS] Done